# Generating Correct Answers for Progressive Matrices Intelligence Tests

**Niv Pekar**
Tel Aviv University
nivpekar@mail.tau.ac.il

**Yaniv Benny**
Tel Aviv University
yanivbenny@mail.tau.ac.il

**Lior Wolf**
Facebook AI Research
and Tel Aviv University

## Abstract

Raven's Progressive Matrices are multiple-choice intelligence tests, where one tries to complete the missing location in a $3 \times 3$ grid of abstract images. Previous attempts to address this test have focused solely on selecting the right answer out of the multiple choices. In this work, we focus, instead, on generating a correct answer given the grid, without seeing the choices, which is a harder task, by definition. The proposed neural model combines multiple advances in generative models, including employing multiple pathways through the same network, using the reparameterization trick along two pathways to make their encoding compatible, a dynamic application of variational losses, and a complex perceptual loss that is coupled with a selective backpropagation procedure. Our algorithm is able not only to generate a set of plausible answers, but also to be competitive to the state of the art methods in multiple-choice tests.

## 1  Introduction

Multiple choice questions provide the examinee with the ability to compare the answers, in order to eliminate some choices, or even guess the correct one. Even when validating the choices one by one, the examinee can benefit from comparing each choice with the query and infer patterns that would have been missed otherwise. Indeed, it is the ability to synthesize de-novo answers from the space of correct answers that is the ultimate test for the understanding of the question.

In this work, we consider the task of generating a correct answer to a Raven Progressive Matrix (RPM) type of intelligence test [8, 2]. Each query (a single problem) consists of eight images placed on a grid of size $3 \times 3$. The task is to generate the missing ninth image, which is on the third row of the third column, such that it matches the patterns of the rows and columns of the grid.

The method we developed has some similarities to previous methods that recognize the correct answer out of eight possible choices. These include encoding each image and aggregating these encodings along rows and columns. However, the synthesis problem demands a new set of solutions.

Our architecture combines three different pathways: reconstruction, recognition, and generation. The reconstruction pathway provides supervision, that is more accessible to the network when starting to train, than the other two pathways, which are much more semantic. The recognition pathway shapes the representation in a way that makes the semantic information more explicit. The generation pathway, which is the most challenging one, relies on the embedding of the visual representation from the first task, and on the semantic embedding obtained with the assistance of the second, and maps the semantic representation of a given query to an image.

In the intersection of the reconstruction and the generation pathways, two embedding distributions need to become compatible. This is done through variational loss terms on the two paths. Since there are many different answers for every query, the variational formulation is also used to introduce randomness. However, since the representation obtained from the generation task is conditioned on a

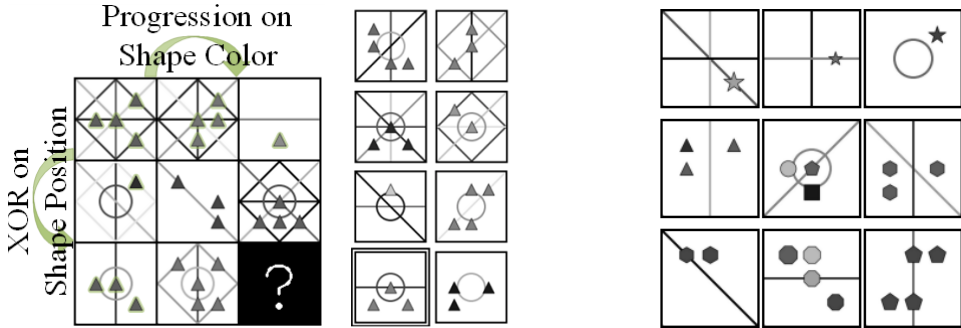

Figure 1: Example of a PGM problem with a $3 \times 3$ grid and 8 choices.

Figure 2: Each row has equivalent answers for some attribute (from top: shape type, shape number, and line color)

complex pattern, uniform randomness can be detrimental. For this reason, we present a new form of a variational loss, which varies dynamically for each condition, to support partial randomness.

Due to the non-deterministic nature of the problem and to the high-level reasoning required, the generation pathway necessitates a semantic loss. For this reason, we employ a perceptual loss that is based on the learned embedding networks. Since the suitability of the generated image is only exposed in the context of the row and column to which it belongs, the perceptual representation requires a hierarchical encoding.

Given the perceptual representation, a contrastive loss is used to compare the generated image to both the correct, ground truth, choice images, as well as to the other seven distractors. Since the networks that encode the images and subsequently the rows and columns also define the context embedding for the image generation, backpropogation needs to be applied with care. Otherwise, the networks that define the perceptual loss would adapt in order to reduce the loss, instead of the generating pathway that we wish to improve.

Our method presents very convincing generation results. The state of the art recognition methods regard the generated answer as the right one in a probability that approaches that of the ground truth answer. This is despite the non-deterministic nature of the problem, which means that the generated answer is often completely different pixel-wise from the ground truth image. In addition, we demonstrate that the generation capability captures most rules, with little neglect of specific ones. Finally, the recognition network, which is employed to provide an auxiliary loss, is almost as effective as the state of the art recognition methods.

## 2 Related work

In RPM, the participant is given the first eight images of a $3 \times 3$ grid (context images) and has to select the missing ninth image out of a set of eight options (choice images). The correct image fits the most patterns over the rows and columns of the grid. In this work, we utilize two datasets: (i) the Procedurally Generated Matrices (PGM) [10] dataset, which depicts various lines and shapes of different types, colors, sizes and positions. Each grid has between one and four rules on both rows and columns, where each rule applies to either the shapes or the lines, see Fig. 1 for a sample, and (ii) the recently proposed RAVEN-FAIR [1] dataset, which is a RPM dataset that is based on the RAVEN [13] dataset and is aimed to remove a bias in the process of creating the seven distractors.

In parallel to presenting PGM, Santoro et al. presented the Wild Relation Network (WReN) [10], which considers the choices one by one. Considering all possible answers at once, both CoPINet [14] and LEN [15] were able to infer more context about the question. CoPINet and LEN also introduced row-wise and column-wise relation operations. MRNet [1] introduced multi-scale processing of the eight query and eight challenge images. Another contribution of MRNet is a new pooling technique for aggregating the information along the rows and the columns of the query's grid. Variational autoencoders [6] were considered as a way to disentangle the representations for improving on held-out rules [11].

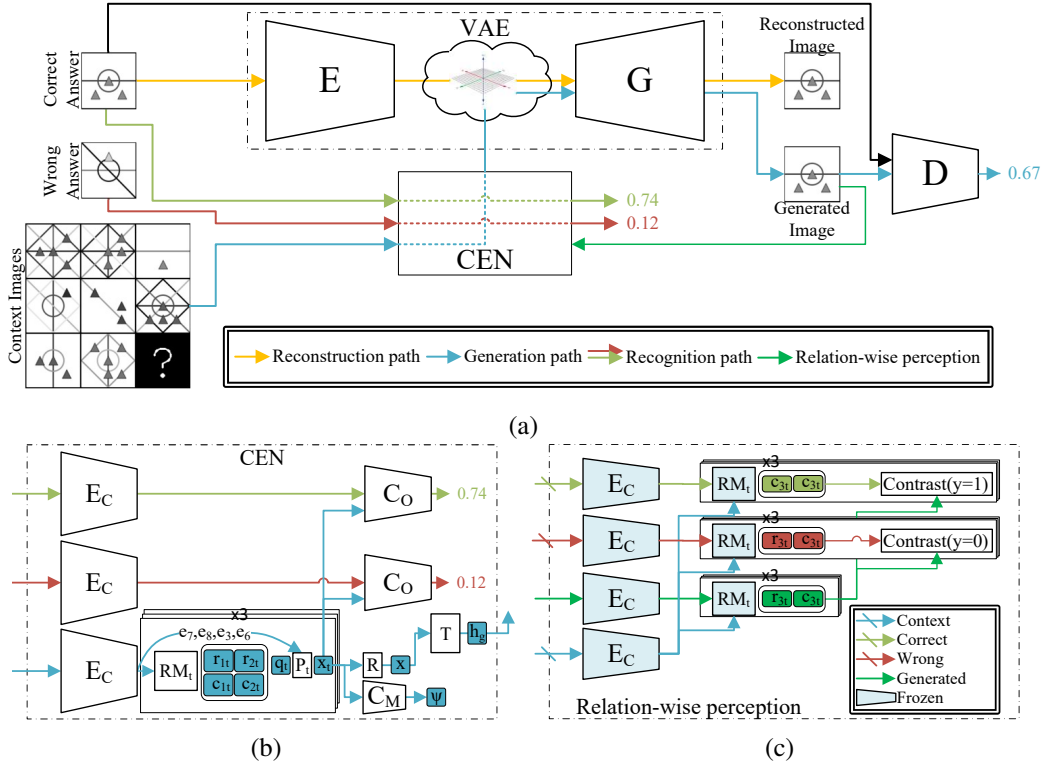

Figure 3: (a) Our architecture. (b) The CEN part. (c) Application of the relation-wise loss.

While our architecture uses a similar pooling operator as MRNet, unlike [1, 14, 15], we cannot perform pooling that involves the third row and the third column. Otherwise, information from the choices would leak to our generator. Despite having this limitation, the ability of our method to select the correct answer does not fall short of any of the previous methods, except for MRNet.

Our work is related to supervised image-to-image translation [4, 12], since the input takes the form of a stack of images. However, the semantic nature of the problem requires a much more abstract representation. For capturing high-level information, the perceptual loss is often used following Johnson et al. [5]. Closely related to it is the usage of feature matching loss terms when training GANs [9]. In this case, one uses the trained discriminator to provide a semantic feature map for matching a distribution, or, in the conditional generation case, to compare the generated image to the desired output. In our work, we use an elaborate type of a perceptual loss, using some of the trained networks, in order to provide a training signal based on the query's rows and columns.

## 3 Method

Our method consists of the following main components: (i) an encoder $E$ and (ii) a generator $G$, which are trained together as a variational autoencoder (VAE) on the images; (iii) a Context Embedding Network (CEN), which encodes the context images and produces the embedding for the generated answer, and (iv) a discriminator $D$, which provides an adversarial training signal for the generator. An illustration of the model can be found in Fig. 3(a). In this section, we describe the functionality of each component. The exact architecture of each one is listed in the supplementary.

**Variational autoencoder** The VAE pathway contains the encoder $E$ and the generator $G$, and it autoencodes the choice images $\{I_{a_i} | i \in [1, 8]\}$ one image at a time. The latent vector produced by the encoder is sampled with the reparameterization trick [6] with a random vector $z$ sampled from a unit gaussian distribution.

$$\mu_v^{a_i}, \sigma_v^{a_i} = E(I_{a_i}) \quad h_{a_i} = \mu_v^{a_i} + \sigma_v^{a_i} \circ z \quad \hat{I}_{a_i} = G(h_{a_i}), \tag{1}$$

where $\mu_v^{a_i}, \sigma_v^{a_i}, h_{a_i}, z \in \mathbb{R}^{64}$, and $\circ$ marks the element-wise multiplication. (The symbol $v$ is used to distinguish from the second reparameterization trick employed by the translation pathway below, with the symbol $g$). The VAE is trained with the following loss $\mathcal{L}_{VAE} = \frac{1}{8} \sum_{i=1}^{8} \left( \lambda_{KL_1} \cdot \mathcal{D}_{KL}(\mu_v^{a_i}, \sigma_v^{a_i}) + MSE(\hat{I}_{a_i}, I_{a_i}) \right)$, where $\mathcal{D}_{KL}$ is the KL-divergence between $h_{a_i}$ and a unit gaussian distribution, $MSE$ is the mean squared error on the image reconstruction and $\lambda_{KL_1} = 4$ is a tradeoff parameter between the KL-Divergence and the reconstruction terms.

**Context embedding network**   Our CEN is comprised of multiple sub-modules, which are trained together to provide input to the generator, see Fig. 3(b) for illustration. The network has an additional auxiliary task of predicting if a choice image is a correct answer conditioned on the context images and predicting an auxiliary 12-bit "meta data" ($\psi$) that define the applied rules.

Our embedding process has some similarity to previous work, MRNet [1]. Our method follows the same encoding process, where it relies on a multi-stage encoder to encode the images into multiple resolutions and relies on a similar Reasoning Module (RM). The key difference from MRNet is that the generation pathway cannot be allowed to observe the choice images $I_{a_i}$, since it will then learn to retrieve the correct image instead of fully generating it.

To generate the answer image, we first encode each context image $I_i$, $i = 1, 2, .., 8$ with the mutli-scale context encoder $(E_h^C, E_m^C, E_l^C)$ in order to extract encodings in three different scales $e_h^i \in \mathbb{R}^{64,20,20}, e_m^i \in \mathbb{R}^{128,5,5}, e_l^i \in \mathbb{R}^{256,1,1}$. These are applied sequentially, such that the tensor output of each scale is passed to the encoder of the following scale.

$$e_h^i = E_h^C(I_i), \quad e_m^i = E_m^C(e_h^i), \quad e_l^i = E_l^C(e_m^i) \tag{2}$$

Since the context embedding is optimized with an auxiliary classifier $C$ that predicts the correctness of each choice, the encoder also encodes the choice images $\{I_{a_i} | i \in [1, 8]\}$, obtaining $\{e_t^{a_i}\}$ for $t \in \{h, m, l\}$. The embeddings of the context images are then passed to a Reasoning Module (RM) in order to detect a pattern between them that will be used to define the properties of the generated image. Since the rules are applied in a row-wise and column-wise orientation, the RM aligns the context embeddings in rows and column, similar to [15, 14, 1]. Following [1], this is done for all three scales, and is denoted by the scale index $t \in \{h, m, l\}$.

The row representations are the concatenated triplets $(e_t^1, e_t^2, e_t^3), (e_t^4, e_t^5, e_t^6)$ and the column representations are $(e_t^1, e_t^4, e_t^7), (e_t^2, e_t^5, e_t^8)$, where the images are arranges as in Fig. 3(a) "Context Images". Each representation is passed through the scale-approriate RM to produce a single representation of the triplet. There is a single RM per scale that encodes both rows and columns.

$$r_t^1 = RM_t(e_t^1, e_t^2, e_t^3), \ r_t^2 = RM_t(e_t^4, e_t^5, e_t^6), \ c_t^1 = RM_t(e_t^1, e_t^4, e_t^7), \ c_t^2 = RM_t(e_t^2, e_t^5, e_t^8) \tag{3}$$

Note that unlike [1], only two rows are used and we do not use the triplets $(e_t^7, e_t^8, e_t^{a_i}), (e_t^3, e_t^6, e_t^{a_i})$, since they contain the embeddings of the choices and using the choices at this stage will reveal the potential correct image to the generation path.

The two row-representations and two column-representations are then joined to form the intermediate context embedding $q_t$. Following [1], this combination is based on the element-wise differences between the vectors, which are squared and summed element-wise. Unlike [1], in our case there are only two rows and two columns:

$$q_t = (r_t^1 - r_t^2)^{.2} + (c_t^1 - c_t^2)^{.2} \tag{4}$$

While the first steps of the CEN shared some of the architecture of previous work, the rest of it is entirely novel. The next step considers the representation of the third row and column in each scale, where the embedding $q_t$ replaces the role of the missing element.

$$x_t = P_t(e_t^7, e_t^8, q_t) + P_t(e_t^3, e_t^6, q_t), \tag{5}$$

where $\{P_t | t \in \{h, m, l\}\}$ is another set of learned sub-networks of the CEN module.

This is the point where the model splits into generation path and recognition path. The merged representations $x_h, x_m, x_l$, are used for two purposes. First, they are used in the auxiliary task that predicts the correctness of each image and the rule type. Second, they are merged to produce a context embedding for the downstream generation.

For the auxiliary tasks, two classifier networks are used, $C_O$ for predicting the correctness of each choice image $I_{a_i}$, and $C_M$ for predicting the rule described in the metadata. The first classifier, unlike

the second, is, therefore, conditioned on the three embeddings $e_t^{a_i}$ of $I_{a_i}$.

$$\hat{y}_i = Sigmoid(C_O(x_h, x_m, x_l, e_h^{a_i}, e_m^{a_i}, e_l^{a_i})), \quad \hat{\psi} = Sigmoid(C_M(x_h, x_m, x_l)), \quad (6)$$

where $\hat{y}_i \in [0, 1]$ and $\hat{\psi} \in [0, 1]^K$, with $K = 12$.

The classifiers apply a binary cross-entropy on the eight choices separately, with $y_i \in \{0, 1\}$ and on the meta target $\psi \in \{0, 1\}^K$.

$$\mathcal{L}_C = \frac{1}{8} \sum_{i=1}^{8} BCE(\hat{y}_i, y_i) + \frac{1}{K} \sum_{k=1}^{K} BCE(\hat{\psi}[k], \psi[k]) \quad (7)$$

For generation purposes, the 3 embeddings $x_h, x_m, x_l$ are combined to a single context embedding $x$

$$x = R(x_h, x_m, x_l), \quad (8)$$

where R is a learned network.

**Generating a plausible answer**   Network $T$ needs to transform the context embedding vector $x$ to a vector in the latent space of the VAE. It maps $x$ to the mean and the standard deviation of a diagonal multivariate Gaussian distribution with parameters $\mu_g, \sigma_g \in \mathbb{R}^{64}$. These are then used together with a random vector $z' \sim \mathcal{N}(0, 1)^{64}$ to sample a new non-deterministic representation $h_g$ in the latent space of the VAE.

$$\mu_g, \sigma_g = T(x) \qquad h_g = \mu_g + \sigma_g \cdot z', \quad (9)$$

Instead of regularizing the reparameterization with the standard KL-divergence loss, we use a novel loss for the reparameterization we call Dynamic Selective KLD (DS-KLD). The loss applies the regularization on a subset of indices, and have this subset change for each case. This way, the model is allowed to reduce the noise on some indices, while other elements of $x$ maintain their information. In other words, we use this novel loss to encourage the model to add noise only for those indices that affect the distracting attributes. This way, it adds variability to the generation process, while not harming the correctness of the generated image. The KL-divergence loss between some i.i.d Gaussian distribution and the normal distribution is defined as two unrelated terms: the mean and the variance:

$$\mathcal{L}_{KL} = -\frac{1}{2} \underbrace{\sum \mu^2}_{\mathcal{L}_{KL_\mu}} -\frac{1}{2} \underbrace{\sum (\log(\sigma^2) - \sigma^2)}_{\mathcal{L}_{KL_\sigma}} \quad (10)$$

Our method applied the mean term as usual, to densely pack the latent space, but applies the variance term only on the subset of indices with variance above the median.

We then use the generator $G$ to synthesize the image.

$$I_g = G(h_g) \quad (11)$$

Two loss terms are applied to the generated image. An unconditional adversarial loss $\mathcal{L}_G$, which trains the generation to produce images that look real, and a conditioned perceptual loss $\mathcal{L}_{COND}$, which trains the generation to produce images with attribute that match the correct answer.

The adversarial loss is optimized with an unconditioned discriminator $D$, which is trained with the standard GAN loss $\min_G \max_D \mathbb{E}_x[\log(D(x))] + \mathbb{E}_z[\log(1 - D(G(z)))]$.

The loss on the discriminator is: $\mathcal{L}_D = \log(D(I_{a^*})) + \log(1 - D(I_g))$, where $a^*$ is the index of the correct target image for the context the generation is conditioned on. The adversarial loss on the Generator (and upstream computations) is $\mathcal{L}_G = \log(D(I_g))$. In order to enforce the generation to be conditioned on the context, we apply a contrastive loss between the generated image and the choices $I_a^i$. For two vectors $x_0, x_1, y \in \{0, 1\}$, and a margin hyper-parameter $\alpha$, the contrastive loss is defined as: $\text{Contrast}(x_0, x_1, y) := y \cdot \|x_0 - x_1\|_2^2 + (1 - y) \cdot \max(0, \alpha - \|x_0 - x_1\|_2^2)$. This loss learns a metric between the two vectors given that they are of the same type ($y = 1$) or not ($y = 0$).

Measuring similarity in this setting is highly nontrivial. The images can be compared on the pixel-level (directly comparing the images) or with a perceptual loss on some semantic level (comparing the image encodings $e_t^i$). However, two images can be very different pixel-wise and semantic-wise and still both be correct with respect to the relational rules. This is shown in Fig. 2, where each of the three images would be considered correct under the specified rule, but would not be correct under any

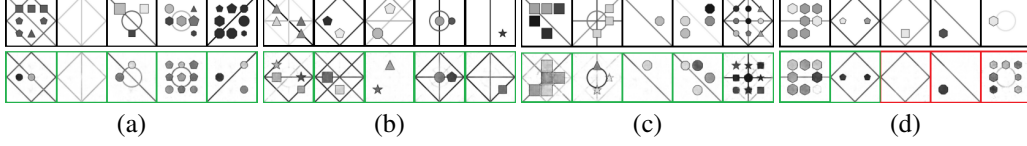

Figure 4: Generation results for selected rules in PGM, each with 5 problems. The top row is the ground truth answer. The bottom is the generated. The good (bad) results are highlighted in green (red) respectively. (a) line type. (b) shape position. (c) shape number. (d) shape type.

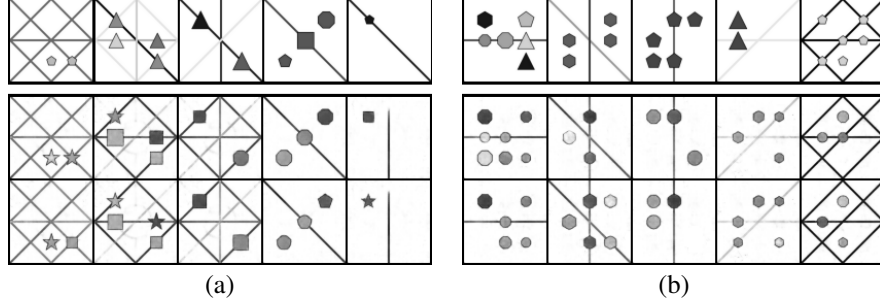

Figure 5: Generation variability of distracting attributes by sampling $z'$ (Eq. 9). A collection of five different PGM problems. the first row contains the real answers and the other rows contain two different generated answers from the same context and different $z'$. (a) shape position. (b) line type.

other rule. For this reason, we do not follow any of these approaches. Our approach is to apply the perceptual loss, conditioned on the context, by computing the third row and column representations $(e_t^7, e_t^8, e_t^9), (e_t^3, e_t^6, e_t^9)$, where $e_t^9 \in \{e_t^g\} \cup \{e_t^{a_i}|i \in [1,8]\}$, compute their relation-wise encodings, and compare those of the generated image to those of the choice images. Here, the index $a_i$ is used to denote the embedding that arises when $I_a = I_{a_i}$, and $e_t^g$ is the encoding of the generated image $I_g$.

The relation-wise encodings are formulated as: $\dot{e}_h^g = \dot{E}_h^C(I_g), \dot{e}_m^g = \dot{E}_m^C(\dot{e}_h^g), \dot{e}_l^g = \dot{E}_l^C(\dot{e}_m^g), r_t^{3,g} = \dot{RM}_t(\ddot{e}_t^7, \ddot{e}_t^8, \dot{e}_t^g), c_t^{3,g} = \dot{RM}_t(\ddot{e}_t^3, \ddot{e}_t^6, \dot{e}_t^g), r_t^{3,a_i} = \dot{RM}_t(\ddot{e}_t^7, \ddot{e}_t^8, \ddot{e}_t^{a_i}), c_t^{3,a_i} = \dot{RM}_t(\ddot{e}_t^3, \ddot{e}_t^6, \ddot{e}_t^{a_i})$. Here, $\ddot{e}$ means that the variable does not backpropagate (a detached copy). $\dot{RM}_t$ specifies that this network is frozen as well. The single dot $\dot{e}$ means that the encoders $E_h^C, E_m^C, E_l^C$ were frozen for this encoding. The rest of the modules ,$G, T, R, P_t$, along with $E_t^C, RM_t$ (through their first paths only), which are part of the generation of $I_g$, are all trained through this loss. This selective freezing of the model is done, since the frozen model is used as a critic in this instance and one cannot optimize $E_t, RM_t$ to artificially try to reduce this loss.

The total contrastive loss $\mathcal{L}_T$ is applied by computing the contrastive loss between the generated image and the choice images $I_{a_i}, i \in [1,8]$.

$$\mathcal{L}_{T,r}^{1^t} = \text{Contrast}(r_t^{3,g}, r_t^{3,a^*}, 1), \quad \mathcal{L}_{T,r}^{0^t} = \frac{1}{7} \sum_{i:a_i \neq a^*} \left( \text{Contrast}(r_t^{3,g}, r_t^{3,a_i}, 0) \right)$$

$$\mathcal{L}_{T,c}^{1^t} = \text{Contrast}(c_t^{3,g}, c_t^{3,a^*}, 1), \quad \mathcal{L}_{T,c}^{0^t} = \frac{1}{7} \sum_{i:a_i \neq a^*} \left( \text{Contrast}(c_t^{3,g}, c_t^{3,a_i}, 0) \right)$$

$$\mathcal{L}_T^{1^t} = \mathcal{L}_{T,r}^{1^t} + \mathcal{L}_{T,c}^{1^t}, \quad \mathcal{L}_T^{0^t} = \mathcal{L}_{T,r}^{0^t} + \mathcal{L}_{T,c}^{0^t}$$

$$\mathcal{L}_T^1 = \mathcal{L}_T^{1^h} + \mathcal{L}_T^{1^m} + \mathcal{L}_T^{1^l}, \quad \mathcal{L}_T^0 = \mathcal{L}_T^{0^h} + \mathcal{L}_T^{0^m} + \mathcal{L}_T^{0^l}$$

$$\mathcal{L}_{COND} = \mathcal{L}_T^1 + \mathcal{L}_T^0$$

(12)

In the ablation, we apply other variants of this loss. (1) Contrastive pixel-wise comparison to $I_a$: $\mathcal{L}_{COND_1} = Contrast(I_g, I_{a^*}, 1) + \frac{1}{7} \sum_{i:a_i \neq a^*} (\text{Contrast}(I_g, I_{a_i}, 0))$, (2) Contrastivefeature-wise comparison to $e^a$: $\mathcal{L}_{COND_2} = \text{Contrast}(\dot{e}^g, \ddot{e}^{a^*}, 1) + \frac{1}{7} \sum_{i:a_i \neq a^*} (Contrast(\dot{e}^g, \ddot{e}^{a_i}, 0))$, and (3) Non-contrastive, without $\mathcal{L}_T^0$ (MSE): $\mathcal{L}_{COND_3} = \mathcal{L}_T^1$.

# 4 Experiments

The Adam optimizer is used with a learning rate of $10^{-4}$. The margin hyper-parameter $\alpha$ (for the contrastive loss) is updated every 1000 iterations to be the mean measured distance between the choices images and the generated. The contrastive loss with respect to the target choice image was multiplied by $3 \cdot 10^{-3}$, and the contrastive loss with respect to the negative choice image was multiplied by $10^{-4}$. The VAE losses were multiplied by 0.1 (with $\beta$ of 4), and the auxiliary $C_m$ loss was multiplied by 10. all other losses were not weighted. The CEN was trained for 5 epochs for the recognition pathway only, after which all subnetworks were trained for ten additional epochs. We train on the train set and evaluate on the test set for all datasets.

The experiments were conducted on the two regimes of the PGM dataset [10], "neutral" and "interpolation" as well as on the recently proposed RAVEN-FAIR dataset [1]. In "neutral" and RAVEN-FAIR, train and test sets are of the same distribution, and in "interpolation", ordered attributes (colour and size) differ in test and train sets. In order to evaluate the generated results, two different approaches were used: machine evaluation (using other recognition models), and human evaluation.

**Machine evaluation in the "neutral" regime of PGM**    A successful generation would present two properties: (i) image quality would be high and the generated images would resemble the ground truth images of the test set. (ii) the generated answers would be correct. The first property is evaluated with FID [3] that is based on an PGM classification networks (see supplementary). To evaluate generation correctness, we employ the same automatic recognition networks. While these networks are not perfectly accurate they support reproducibility. To minimize bias, both evaluations are repeated with three pretrained models of largely different architectures: WReN [10], LEN [15] and MRNet [1].

The accuracy evaluation is performed by measuring the fraction of times, in which the generated answer is chosen over the seven distractors (false choices) of each challenge. This number is compared to the recognition results of each network given the ground truth target $I_{a^*}$. Ideally, our generation method would obtain the same accuracy. However, just by applying reconstruction to $I_{a^*}$, there is a degradation in quality that reduces the reported accuracy. Therefore, to quantify this domain gap between synthetic and real images, we also compare to two other versions of the ground truth: in one it is reconstructed without any randomness added and in the other, no reparameterization is applied. These two are denoted by $G(\mu_{a^*}^v)$ and $G(h_{a^*})$, respectively.

As can be seen in Table 1, our method, denoted 'Full', performs somewhat lower in terms of accuracy in comparison to the real targets. However, most of this gap arises from the synthetic to real domain gap. It is also evident that this gap is larger when randomness is added to the encoding before reconstruction takes place. However, for our method (comparing it to 'Full, w/o reparam in test') this gap is smaller, suggesting it was adapted for this randomization.

Considering the FID scores, we can observe that while the FID of the generated answer is somewhat larger than that of the reconstructed versions of the ground truth answer, it is still relatively low. This is despite the VAE itself lacking as a generator for random seeds, as is evident when considering the 'random VAE image' row in the table. This row is obtained by sampling from the normal distribution in the latent space of the VAE and performing reconstruction. Autoencoders, unlike GANs, usually do not provide good reconstruction for random seeds.

**Generated examples**    The visual quality of the output can be observed by considering the examples in Fig. 4. It is evident that the generated images mimic the domain images, yet, maybe with a tendency for fainter colors. More images are shown in the supplementary.

The generations in Fig. 4 are in the context of a specific query that demonstrates a selected rule. The top row shows the ground truth answer, and the bottom row shows the generated one. The correct answers (validated manually) are marked in green. As can be seen, the correct solution generated greatly differs from the ground truth one, demonstrating the variability in the space of correct answers. This variability is also demonstrated in Fig. 5, in which we present two solutions (out of many) for a given query. The obtained solutions are different. However, in some cases, they tend to be more similar to one another than to the ground truth answer.

**Ablation study in the "neutral" regime of PGM**    To validate the contribution of each of the major components of our method, an ablation analysis was conducted. The following variations of our full method are trained from scratch and tested: (1) **W/o reparameterization in train:** generate answers

Table 1: Performance on each evaluator. Acc is $I_g$ vs. the seven $I_a$ for $a \neq a*$.

| | WReN | | LEN | | MRNet | |
|---|---|---|---|---|---|---|
| | Acc | FID | Acc | FID | Acc | FID |
| Real Target ($I_{a*}$) | 76.9 | - | 79.6 | - | 93.3 | - |
| Recon. Target ($G(\mu_{a*}^v)$) | 62.9 | 1.9 | 66.2 | 12.2 | 80.6 | 2.9 |
| Recon. Target with reparam ($G(h_{a*})$) | 58.4 | 2.2 | 62.3 | 14.2 | 76.5 | 3.6 |
| Full | 58.7 | 5.9 | 60.1 | 38.6 | 65.4 | 8.1 |
| Full, w/o reparam. in test | **59.0** | 4.9 | **60.4** | 37.3 | **65.5** | 7.5 |
| (1) W/o reparam. in train | 54.7 | **4.7** | 56.3 | **37.2** | 60.1 | **7.1** |
| (2) Standard KLD | 47.4 | 5.7 | 49.3 | 38.1 | 53.3 | 7.6 |
| (3) Static half KLD | 50.6 | 6.0 | 52.9 | 40.2 | 55.8 | 8.8 |
| (4) W/o VAE | 50.5 | 6.2 | 52.7 | 44.0 | 55.6 | 12.7 |
| (5) W/o auxiliary $C_M$ | 53.6 | 6.2 | 54.7 | 38.8 | 57.2 | 8.2 |
| (6) W/o $\mathcal{L}_T$ | 51.5 | 5.2 | 50.4 | 40.9 | 52.7 | 8.5 |
| (7) $\mathcal{L}_T$ on $e^a$ | 47.2 | 6.8 | 46.6 | 41.3 | 48.4 | 8.4 |
| (8) $\mathcal{L}_T$ on $I_a$ | 46.0 | 8.0 | 46.7 | 613.9 | 48.1 | 65.0 |
| (9) W/o freeze | 40.3 | 7.5 | 44.8 | 869.0 | 44.6 | 58.1 |
| Random VAE image | 22.3 | 8.1 | 29.6 | 1637.0 | 21.5 | 94.4 |

Table 2: Accuracy and FID per rule (MRNet)

| | Line | | Shape | | | | |
|---|---|---|---|---|---|---|---|
| Model | Type | Color | Type | Color | Pos. | Num. | Size |
| Acc. on Real Target | 96.3 | 96.3 | 88.3 | 76.5 | 99.0 | 98.5 | 89.4 |
| Acc. on recon. Target | 89.8 | 83.1 | 75.6 | 60.4 | 90.1 | 81.2 | 73.4 |
| Acc. of our method | 86.6 | 57.2 | 41.0 | 37.8 | 88.0 | 55.9 | 45.7 |
| FID of our method | 4.9 | 6.52 | 5.3 | 6.1 | 5.0 | 6.4 | 6.1 |

Table 3: The accuracy obtained by the auxiliary classifier $C_o$

| Model | PGM | PGM_aux |
|---|---|---|
| WReN [10] | 62.6 | 76.9 |
| CoPINet [14] | 56.4 | - |
| LEN [15] | 68.1 | 82.3 |
| MRNet [1] | 93.3 | 92.6 |
| Our $C_o$ | 68.2 | 82.9 |

without the reparameterization trick, but use a random vector $z' \sim \mathcal{N}(0,1)^{128}$ concatenated to the context embedding. (2) **Standard KLD:** reparameterization trick employing the standard KLD loss. (3) **Static half KLD:** applying the KLD loss to a fixed half of the latent space and do not apply it to the other indices. (4) **W/o $VAE$:** without autoencoding, using a discriminator to train a GAN on the generated images and the real answer images. (5) **W/o auxiliary $C_M$:** without the auxiliary task of predicting the correct rule type. (6) **W/o $\mathcal{L}_T$:** instead of using contrastive loss $\mathcal{L}_T$, we trained using the MSE loss just for minimizing the relation distance between the generated image to the target image. (7) $\mathcal{L}_T$ **on $e^a$:** instead of using contrastive perceptual loss with the relation module, we train by using contrastive feature-wise loss with the features encoded vectors $e_a$: $\mathcal{L}_{COND_2} = \text{Contrast}(\dot{e}^g, \ddot{e}^{a^*}, 1) + \frac{1}{7}\sum_{i:a_i \neq a*}(\text{Contrast}(\dot{e}^g, \ddot{e}^{a_i}, 0))$. (8) $\mathcal{L}_T$ **on $I_a$:** instead of using contrastive perceptual loss with the relation module, we train by using contrastive pixel-wise comparison loss on the images $I_a$: $\mathcal{L}_{COND_1} = \text{Contrast}(I_g, I_{a*}, 1) + \frac{1}{7}\sum_{i:a_i \neq a*}(\text{Contrast}(I_g, I_{a_i}, 0))$. Finally, (9) **W/o freeze:** without selectively freezing of the model. Variants 1–4 test the selective reparameterization and the VAE, variants 6-9 test the perceptual loss and its application.

As can be seen in Table 1, each of these variants leads to a decrease in the accuracy of the generated answer performance. For FID the effect is less conclusive and there is often a trade-off between it and the accuracy. Interestingly, removing randomness altogether is better than using the conventional KLD term, in which there is no selection of half of the vector elements.

**Performance on each task in the "neutral" regime of PGM** Generative models often suffer from mode collapse and even if, on average, the generation achieves high performance, it is important to evaluate the success on each rule. For this purpose, we employ MRNet, which is currently the most accurate classification model. In Tab. 2 we present the breakdown of the accuracy per type of rule. As can be seen, the performance of our generated result is not uniformly high. There are rules such as Line-Type and Shape-Pos that work very well, and rules such as Shape-Type and Shape-Color that our method struggles with.

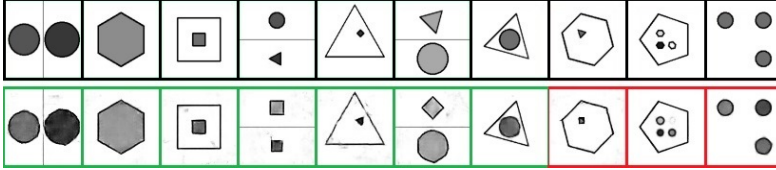

Figure 6: A collection of ten different RAVEN-FAIR problems. Real target images on the top, and generated images on the bottom. some attributes are allowed to change when no rules are applied on them (correct in green, incorrect in red).

Table 4: Recognition results on RAVEN-FAIR

| Model | Accuracy |
|---|---|
| WReN [10] | 30.3 |
| CoPINet [14] | 50.6 |
| LEN [15] | 51.0 |
| MRNet [1] | 86.8 |
| Our $C_o$ | 60.8 |

**Human evaluation in the "neutral" regime of PGM**   Two user studies were conducted. The first is a user study that follows the same scheme as the machine evaluation. Three participants were extensively trained on the task of PGM questions. After training, each got 30 random questions with the correct target image, reconstructed by VAE to match the quality, and 30 with the generated target instead. Human performance on the correct image was 72.2%, and on the generated image was 63.3%. We note that due to the extensive training required, the number of participants leads to a small sample size.

To circumvent the training requirement, we conducted a second user that is suitable for untrained individuals. The study is motivated by the qualitative image analysis in Fig. 4. In PGM, an image is correct if and only if it contains the right instance of the object attribute which the rule is applied on (this information is in the metadata). By comparing the generated object attribute to the correct answer object attribute, one can be easily determined if the generation is correct. This study had $n = 22$ participants, 140 random image comparing instances for the generated answers, and 140 for a random choice image (reconstructed by VAE) as a baseline. The results show that 70.1% of the generations were found to be correct and only 6.4% of the random choice images (baseline).

**Recognition performance in the "neutral" regime of PGM**   We evaluate the recognition pathway, i.e., the accuracy obtained by the classifier $C_O$. This classifier was trained as an auxiliary classifier, and was designed with the specific constraint of not using the relational module RM on the third row and column. It is therefore at a disadvantage in comparison to the literature methods. Tab. 3 presents results for two versions of $C_O$. One was trained without the metadata (this is the 'W/o auxiliary $C_M$ ablation') and one with. These are evaluated in comparison to classifiers that were trained with and without this auxiliary information. As can be seen, our method is highly competitive and is second only to MRNet [1].

**The "interpolation" regime of PGM**   Out-of-distribution generalization was demonstrated by training on this regime of PGM, in which the ordered attributes (colour and size) differ in test and train sets. Evaluation was done using the MRNet model that was trained on the "neutral" regime. The generation accuracy was 61.7%, which is very close to the 68.1% (MRNet) and 64.4% (WReN) accuracy in the much easier recognition task in this regime.

**"RAVEN-FAIR"**   Further experiments were done on this recent variant of the RAVEN dataset, see Fig. 6 for some typical examples. machine evaluation was performed using MRNet, which is the state of the art network for this dataset, with 86.8% accuracy. The generation accuracy was 60.7%, this is to be compared to 69.5% on the target image reconstructed by VAE, and only 8.9% on a random generated image. We also evaluate the recognition pathway (auxiliary classifier $C_O$ performance). Tab. 4 presents those results in comparison to other classifiers. It seems that for RAVEN-FAIR, our method achieve far greater recognition results then most classifiers, and is second only to MRNet [1].

## 5   Conclusions

In problems in which the solution space is complex enough, the ability to generate a correct answer is the ultimate test of understanding the question, since one cannot extract hints from any of the potential answers. Our work is the first to address this task in the context of RPMs. The success in this highly semantic task relies on a large number of crucial technologies: applying the reparameterization trick selectively and multiple times, reusing the same networks for encoding and to provide a loss signal, selective backpropagation, and an adaptive variational loss.

## Broader Impact

The shift from selecting an answer from a closed set to generating an answer could lead to more interpretable methods, since the generated output may reveal information about the underlying inference process. Such networks are, therefore, more useful for validating cognitive models through the implementation of computer models.

The field of answer generation may play a crucial part in automatic tutoring. Ideally, the generated answer would fit the level of the student and allow for automated personalized teaching. Such technologies would play a role in making high-level education accessible to all populations.

## Acknowledgements

This project has received funding from the European Research Council (ERC) under the European Unions Horizon 2020 research and innovation programme (grant ERC CoG 725974).

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
