[Supplementary Material]

# Supplementary Material for: Generating Correct Answers for Progressive Matrices Intelligence Tests

## A  Links for datasets

The Procedurally Generated Matrices (PGM) dataset can be found in `https://github.com/deepmind/abstract-reasoning-matrices` and the RAVEN-FAIR dataset can be found in `https://github.com/yanivbenny/RAVEN_FAIR`.

## B  Calculating FID

FID calculation was performed on the real target image and the generated image. For WReN we used the output of the CNN encoder of [32, 4,4] vector, flattened to a 512 size vector. For MRnet we used the outputs of the perception encoders – low, mid and high features sizes – [256, 1, 1], [128, 5, 5] and [64, 20, 20], and average pooled them to 256, 128 and 64 sized vectors. Than concatenated them to 448 size vector. And for LEN we used the output of the CNN encoder of [32, 4,4] vector, and average pooled to a 32 size vector.

## C  Generated images

In Fig. 1 we present some examples of our method's generated images. For analyzing the generation results in PGM we must understand for each of the rules, what counts as a good generated image. For 'line type' (a) rules, the lines must stay at the same place, but all other attributes may change, here we get great results. For 'shape position' (b), the shape's position must always be the same, and all other attributes may change, including the shape's size, type, color, and the lines. Here we can see that almost all of the results are great. For 'shape number' (c), the shape's number must always be the same, and all other attributes may change, including the shape's size, type, color, position, and the lines. Here also, we get great results. For 'shape type' (d), the shape's type and number must always be the same, and all other attributes may change, including the shape's size, color, position, and the lines. Here we can see that the results are average. For 'shape color' (e), the shape's color and number must always be the same, and all other attributes may change, including the shape's size, type, position, and the lines. Here we can see that the results are also average. For 'shape size' (f), the shape's size and number must always be the same, and all other attributes may change, including the shape's type, position, color, and the lines. Results are average. For 'line color' (g), the line's color must always be the same, and all other attributes may change, including the shape's attributes and the line's type. Here results look good.

(a) line type

(b) shape position

(c) shape number

(d) shape type

(e) shape color

(f) shape size

(g) line color

Figure 1: Generation results for selected rules in PGM, each with 10 problems. The top row is the ground truth answer. The bottom is the generated. (a) line type. (a) shape position. (c) shape number. (d) shape type.(e) shape color. (f) shape size. (g) line color.

# D   Architecture details

We detail each sub-module used in our method in Tab. 2-6. Since some modules re-use the same blocks, Tab. 1 details a set of general modules.

Table 1: ResBlocks, with variable number of channels $c$.

| Module | layers | parameters | input | output |
|---|---|---|---|---|
| ResBlock(c) | Conv2D | CcK3S1P1 | $x$ | |
| | BatchNorm | | | |
| | ReLU | | | |
| | Conv2D | CcK3S1P1 | | |
| | BatchNorm | | | $x'$ |
| | Residual | | $(x, x')$ | $x'' = x + x'$ |
| | ReLU | | | |
| DResBlock(c) | Conv2D | CcK3S1P1 | $x$ | |
| | BatchNorm | | | |
| | ReLU | | | |
| | Conv2D | CcK3S1P1 | | |
| | BatchNorm | | | $x'$ |
| | Conv2D | CcK1S2P0 | $x$ | |
| | BatchNorm | | | $x_d$ |
| | Residual | | $(x_d, x')$ | $x'' = x_d + x'$ |
| | ReLU | | | |
| ResBlock1x1(c) | Conv2D | CcK1S1P1 | $x$ | |
| | BatchNorm | | | |
| | ReLU | | | |
| | Conv2D | CcK1S1P1 | | |
| | BatchNorm | | | $x'$ |
| | Residual | | $(x, x')$ | $x'' = x + x'$ |
| | ReLU | | | |

Table 2: $E^C$ and $RM$ modules

| Module | layers | input | output |
|---|---|---|---|
| $E_h^C$ | Conv2d(1, 32, kernel size=7, stride=2, padding=3, bias=False) | $I_i$ | - |
| | BatchNorm2d(32) | - | - |
| | ReLU | - | - |
| | Conv2d(32, 64, kernel size=3, stride=2, padding=1, bias=False) | - | |
| | BatchNorm2d(64) | - | - |
| | ReLU | - | $e_h^i$ |
| $E_m^C$ | Conv2d(64, 64, kernel size=3, stride=2, padding=1, bias=False) | $e_h^i$ | - |
| | BatchNorm2d(64) | - | - |
| | ReLU | - | - |
| | Conv2d(64, 128, kernel size=3, stride=2, padding=1, bias=False) | - | - |
| | BatchNorm2d(128) | - | - |
| | ReLU | - | $e_m^i$ |
| $E_l^C$ | nn.Conv2d(128, 128, kernel size=3, stride=2, padding=1, bias=False) | $e_m^i$ | - |
| | BatchNorm2d(128) | - | - |
| | ReLU | - | - |
| | Conv2d(128, 256, kernel size=3, stride=2, padding=0, bias=False) | - | - |
| | BatchNorm2d(256) | - | - |
| | ReLU | - | $e_l^i$ |
| $RM_h$ | Reshape to (3 * 64, 20, 20) | - | - |
| | Conv2d(3*64, 64, ker size=3, st=1, pad=1, bias=False) | - | - |
| | ResBlock(64, 64) | - | - |
| | ResBlock(64, 64) | - | - |
| | Conv2d(64, 64, ker size=3, st=1, pad=1, bias=False) | - | - |
| | BatchNorm2d(64) | - | $r_h^1/r_h^2/c_h^1/c_h^2$ |
| $RM_m$ | Reshape to (3 * 128, 5, 5) | - | - |
| | Conv2d(3 * 128, 128, ker size=3, st=1, pad=1, bias=False) | - | - |
| | ResBlock(128, 128) | - | - |
| | ResBlock(128, 128) | - | - |
| | Conv2d(128, 128, ker size=3, st=1, pad=1, bias=False) | - | - |
| | BatchNorm2d(128) | - | $r_m^1/r_m^2/c_m^1/c_m^2$ |
| $RM_l$ | Reshape to (3 * 256, 1, 1) | - | - |
| | Conv2d(3*256, 256, ker size=1, st=1, pad=1, bias=False) | - | - |
| | ResBlock1x1(256, 256) | - | - |
| | ResBlock1x1(256, 256) | - | - |
| | Conv2d(256, 256, ker size=3, st=1, pad=1, bias=False) | - | - |
| | BatchNorm2d(256) | - | $r_l^1/r_l^2/c_l^1/c_l^2$ |

Table 3: $P$ and $C_O$ modules

| Module | layers | input | output |
|---|---|---|---|
| $P_h$ | Reshape to (3 * 64, 20, 20) | $e_h^7, e_h^8, q_h$ /$e_h^3, e_h^6, q_h$ | - |
| | Conv2d(3*64, 64, ker size=3, st=1, pad=1, bias=False) | - | - |
| | ResBlock(64, 64) | - | $x_h$ |
| $P_m$ | Reshape to (3 * 128, 5, 5) | $e_m^7, e_m^8, q_m$ /$e_m^3, e_m^6, q_m$ | - |
| | Conv2d(3*128, 128, ker size=3, st=1, pad=1, bias=False) | - | - |
| | ResBlock(128, 128) | - | $x_m$ |
| $P_l$ | Reshape to (3 * 256, 1, 1) | $e_l^7, e_l^8, q_l$ /$e_l^3, e_l^6, q_l$ | - |
| | Conv2d(3*256, 256, ker size=3, st=1, pad=1, bias=False) | - | - |
| | ResBlock(256, 256) | - | $x_l$ |
| $C_O^h$ | Reshape to (2 * 64, 20, 20) | cat($x_h, e_h^{a_i}$) | - |
| | Conv2d(2*64, 64, ker size=3, st=1, pad=1, bias=False) | - | - |
| | ResBlock(64, 64) | - | - |
| | DResBlock(64,2 *64, stride=2) | - | - |
| | DResBlock(2 *64,128, stride=2) | - | - |
| | AdaptiveAvgPool2d((1, 1)) | - | $x_h'$ |
| $C_O^m$ | Reshape to (2 * 128, 5, 5) | cat($x_m, e_m^{a_i}$) | - |
| | Conv2d(2*128, 128, ker size=3, st=1, pad=1, bias=False) | - | - |
| | ResBlock(128, 128) | - | - |
| | DResBlock(128,2*128, stride=2) | - | - |
| | DResBlock(2*128,128, stride=2) | - | - |
| | AdaptiveAvgPool2d((1, 1)) | - | $x_m'$ |
| $C_O^l$ | Reshape to (3 * 256, 1, 1) | cat($x_l, e_l^{a_i}$) | - |
| | Conv2d(2*256, 256, ker size=3, st=1, pad=1, bias=False) | - | - |
| | ResBlock(256, 256) | - | - |
| | Conv2d(256, 128, ker size=1, st=1, bias=False) | - | - |
| | BatchNorm2d(128) | - | - |
| | ReLU | - | - |
| | ResBlock1x1(128, 128) | - | - |
| | AdaptiveAvgPool2d((1, 1)) | - | $x_l'$ |
| final $C_O$ | Linear(128*3, 256, bias=False) | cat($x_h', x_m', x_l'$) | - |
| | BatchNorm1d(256) | - | - |
| | ReLU | - | - |
| | Linear(256, 128, bias=False) | - | - |
| | BatchNorm1d(128) | - | - |
| | ReLU | - | - |
| | Linear(128, 1, bias=True)) | - | $y_i$ |

Table 4: $C_M$ modules

| Module | layers | input | output |
|---|---|---|---|
| $C_M^h$ | DResBlock(64,2*64, stride=2)<br>DResBlock(2*64,128, stride=2)<br>AdaptiveAvgPool2d((1, 1)) | $x_h$<br>-<br>- | -<br>-<br>$x_h'$ |
| $C_M^m$ | DResBlock(128,2*128, stride=2)<br>DResBlock(2*128,128, stride=2)<br>AdaptiveAvgPool2d((1, 1)) | $x_m$<br>-<br>- | -<br>-<br>$x_m'$ |
| $C_M^l$ | ResBlock(256, 256)<br>Conv2d(256, 128, ker size=1, st=1, bias=False)<br>BatchNorm2d(128)<br>ReLU<br>ResBlock1x1(128, 128)<br>AdaptiveAvgPool2d((1, 1)) | $x_l$<br>-<br>-<br>-<br>-<br>- | -<br>-<br>-<br>-<br>-<br>$x_l'$ |
| $C_M$ | Linear(128*3, 256, bias=False)<br>BatchNorm1d(256)<br>ReLU<br>Linear(256, 128, bias=False)<br>BatchNorm1d(128)<br>ReLU<br>Linear(128, 12, bias=True)) | cat($x_h'$, $x_m'$, $x_l'$)<br>-<br>-<br>-<br>-<br>-<br>- | -<br>-<br>-<br>-<br>-<br>-<br>$\psi$ |

Table 5: $R$ modules

| Module | layers | input | output |
|---|---|---|---|
| $R_h$ | DResBlock(64,2*64, stride=2)<br>DResBlock(2*64,128, stride=2)<br>AdaptiveAvgPool2d((1, 1)) | $x_h$<br>-<br>- | -<br>-<br>$x_h'$ |
| $R_m$ | DResBlock(128,2*128, stride=2)<br>DResBlock(2*128,128, stride=2)<br>AdaptiveAvgPool2d((1, 1)) | $x_m$<br>-<br>- | -<br>-<br>$x_m'$ |
| $R_l$ | Conv2d(256, 128, ker size=1, st=1, bias=False)<br>BatchNorm2d(128)<br>ReLU<br>ResBlock1x1(128, 128)<br>AdaptiveAvgPool2d((1, 1)) | $x_l$<br>-<br>-<br>-<br>- | -<br>-<br>-<br>-<br>$x_l'$ |
| T | Linear(128*3, 128, bias=False)<br>ReLU<br>Linear(128, 128, bias=False)<br>ReLU<br>Linear(128, 128, bias=False)) | x = cat($x_h'$, $x_m'$, $x_l'$)<br>-<br>-<br>-<br>- | -<br>-<br>-<br>-<br>$mu_g$,$sigma_g$ |

Table 6: G, E and D

| Module | layers | input | output |
|---|---|---|---|
| G | ConvTranspose2d(64, 64*8, ker size=5, st=1, pad=0, bias=False) | $h_g = \mu_g + \sigma_g \cdot z'$ | - |
| | BatchNorm2d(64*8) | - | - |
| | ConvTranspose2d(64*8, 64*4, ker size=4, st=2, pad=1, bias=False) | - | - |
| | BatchNorm2d(64*4) | - | - |
| | ConvTranspose2d(64*4, 64*2, ker size=4, st=2, pad=1, bias=False) | - | - |
| | BatchNorm2d(64*2) | - | - |
| | ConvTranspose2d(64*2, 64*1, ker size=4, st=2, pad=1, bias=False) | - | - |
| | BatchNorm2d(64*1) | - | - |
| | ConvTranspose2d(64*1, 1, ker size=4, st=2, pad=1, bias=False) | - | $I_g$ |
| E | Conv2d(1, 32, kernel size=3, stride=2) | $I_{a_i}$ | - |
| | BatchNorm2d(32) | - | - |
| | ReLU | - | - |
| | Conv2d(32, 32, kernel size=3, stride=2) | - | - |
| | BatchNorm2d(32) | - | - |
| | ReLU | - | - |
| | Conv2d(32, 32, kernel size=3, stride=2 | - | - |
| | BatchNorm2d(32) | - | - |
| | ReLU | - | - |
| | Conv2d(32, 32, kernel size=3, stride=2 | - | - |
| | BatchNorm2d(32) | - | - |
| | ReLU | - | - |
| | Linear(32*4*4, 64*2) | - | $\mu_v^{a_i}, \sigma_v^{a_i}$ |
| D | Conv2d(1, 64, ker size=4, st=2, pad=1, bias=False) | $I_{a^*}/I_g$ | - |
| | leaky relu | - | - |
| | Conv2d(64,64*2, ker size=4, st=2, pad=1, bias=False) | - | - |
| | BatchNorm2d(64*2) | - | - |
| | leaky relu | - | - |
| | Conv2d(64*2,64*4, ker size=4, st=2, pad=1, bias=False) | - | - |
| | BatchNorm2d(64*4) | - | - |
| | leaky relu | - | - |
| | Conv2d(64*4,64*8, ker size=4, st=2, pad=1, bias=False) | - | - |
| | BatchNorm2d(64*8) | - | - |
| | leaky relu | - | - |
| | Conv2d(64*8, 1, ker size=4, st=1, pad=0, bias=False) | - | - |
| | leaky relu | - | - |
| | Conv2d(1, 1, ker size=2, st=1, pad=0, bias=False) | - | - |
| | sigmoid | - | D out |