[Reviews · NeurIPS 2020]

Review 1

Summary and Contributions: This paper describes a method to generate the correct choice panels i.e. the answers to the RPM problems. The proposed method consists of 3 modules. (1), VAE for learning the latent representation of the choice panel, (2), CEN for capturing implicit relations in the context panels (3) DIscriminator for improving the generated image. With properly designed loss functions, the latent representation generated by CEN is used to sample the hidden vector for the generative process in the VAE, therefore the proposed method can be used the context panel the generate the choice image according to the given choice panels. The experiments are designed to automatically verify the correctness of the generated answers.

Strengths: The proposed method does provide a different perspective comparing to the previous methods which learn discriminative models instead of generative ones. The way of using the context panels to generate a hidden representation that approximates the latent representation of VAE is interesting and novel in the RPM domain. Making the whole generation process end-to-end is a valid contribution to the community.

Weaknesses: The design of the experiments needs further illustration and justification. First of all, the selected models for recognizing the correct choice are not strong enough in the selected dataset. The accuracy of these models is not trustworthy in this experimental setup. Second, the accuracies of these models on the generated answer are even worse. It is difficult to tell whether the generated results are correct or not. Last and most importantly, the authors should conduct experiments on the RAVEN dataset because there is robust model [13] that reaches over 90% accuracy. Generating the answer on RAVEN with [13] to evaluate the correctness of the generated results would be a much better way to justify the proposed method empirically.

Correctness: There are some misunderstandings of the definition of generative model. A well acceptable definition should be “sample from a latent variable to generate a corresponding signal”. I would recommend not to use the word generative model. The experiments need further justification.

Clarity: It is easy to read the paper.

Relation to Prior Work: The paper does provide literature review to distinguish itself with the SOTA methods.

Reproducibility: Yes

Additional Feedback: Please address my question about the experimental setup. I am open to raise the score if the authors could provide insightful feedback. I have read the rebuttal and the comments from other reviewers. I would like to keep my original rating because the feedback from the authors did not address my problem. It looks like the authors misunderstand my point in the rebuttal. To be more specific. The method to verify the correctness is not trustworthy because (1), the SOTA to choose to discriminate the answer is not strong enough, only 70% acc. (2) In addition, the verification process is not conducted on the problem individually. It is difficult to tell whether the proposed method generates the correct answer panel for the corresponding context. (3) According to (1) and (2), it makes the entire evaluation metric not reliable. The authors misunderstand my comment in the initial review and make irrelevant responses in the rebuttal. Therefore I decided to keep my initial rating.


Review 2

Summary and Contributions: This paper focuses on the progressive matrices intelligence test, which generates the correct answer given some contextual images. The performance of the model on the generated image quality is satisfying and the performance on multiple-choice tests is competitive.

Strengths: S1: the performance of the proposed model is good, which is able to generate both correct and high-quality images given the abstract images. S2: the task is interesting to me. S3: ablation studies are conducted which demonstrates how each module works.

Weaknesses: W1: What is the performance of humans on the multiple-choice test? Does AI outperform humans? W2: it seems that the proposed model just combines some previous works, like VAE and GAN, which have been widely applied in image generation. The only difference is that the task here is abstract image generation instead of natural image generation. W3: which dataset do the authors use in this paper?

Correctness: Yes.

Clarity: Yes.

Relation to Prior Work: Yes.

Reproducibility: Yes

Additional Feedback:


Review 3

Summary and Contributions: This paper proposes a task of learning to synthesis correct answers for Raven Progressive Matrix. Authors argue that it is a more challenging task than existing RPM-style tasks since the metric of generation is defined on a semantic space. That is, the generated results do not have to be exactly the same as the correct answer, but it should capture the correct underlying relation. To tackle this challenge, authors design a model with three pathways for reconstruction, recognition, and generation. For empirical evaluation, authors compare the generated results and the ground-truth correct results with SOTA recognition models, and find the model tested achieve similar performance. Authors also provide qualitative study to justify the learnt generative model can indeed extract the abstract relations from provided samples.

Strengths: The model is very carefully designed, even though the idea itself is intuitive. First put aside the aesthetics of modeling, this model does take quite some efforts to design and train. Since authors present their methods with pretty smooth logic flow, I can only roughly summarize their ideas here. First there is a reconstruction pathway, which is basically a VAE model for the correct answer. It learns the patterns in the pixel-level. Then there is a recognition pathway, which combines the context provided in the sample images (the first two rows/columns of RPM) with the left ones. This pathway inherits the multi-scale abstraction structure from [1], and is trained to extract contextual relations from given images, which is later used as a context-aware discriminator to train the generator. The generator is based on the decoder of VAE, but in the latent variable is sampled from a distribution conditioned on the context. There is also an unconditioned discriminator that provides GAN-style weak supervision. The empirical results seem interesting. I am amazed the model can generate qualitatively correct images. And the ablation study is comprehensive.

Weaknesses: My first concern is that this model seems far from minimalism. Generating correct answer for RPM is an interesting task. But one of the reasons it is interesting to the current AI community is that humans can somehow generate some results correctly without huge amount of training. Although this work demonstrates the possibility of generator that can show some reasoning capability, I highly speculate that this is a distillation from the subnetworks for context extraction, which is trained with strong supervision. There is still a long distance from this model and human brain. The latter one is believed to be designed by nature following minimalism. And the quest for this minimum model is necessary because the task of RPM itself is not applicable to real life scenes. It is the general meta priors we discovered from this task that might be promising for a more complex real world model. Therefore, the proposed model might be of great interest to communities like image translation, but it is not quite to the point of the research in RPM. With that said, there are also quite some empirical designs that are probably not generalizable to other task. For example, the DS-KLD for the regularization on the generator. Also, VAE are always blamed to in the authenticity of generated images. Authors claim that their evaluation scheme that bypasses human study and replaces it with SOTA model is mainly due to the concern of human resources. However, I really doubt the actual feasibility of human study even if adequate human resources are provided. Apparently, humans can somehow figure out the blurry patterns in the generated images. Did author deliberately choose machines to verify their model because of it? Can authors provide an experiment setup with hypothetically sufficient labour? Other weaknesses are listed in correctness part below.

Correctness: In terms of correctness, authors did not introduce how they split training, testing and evaluation sets. Relational models are normally expected to exhibit certain out-of-distribution capability. I'd like to request authors' comment on this.

Clarity: This paper is generally very well written. Authors offer great details of their logic flows when introducing their model. The method section is very informative and easy to follow.

Relation to Prior Work: I think this work covers most of prior works, though there seems to be a skew in the perspective on the essence of RPM-style problems.

Reproducibility: Yes

Additional Feedback: I raised my rating because authors showed significant efforts in addressing my concerns. The design to use VAE to reconstruct image to add blurriness to wrong answer is intriguing. Even though authors do not provide figures for it, I would like to believe the blurriness affects reconstructed answers and generated answers equally. Also, the human study looks interesting. I highly recommend authors to include them in revision. I would also like to see discussion on weakness#1 in revision to make how this model leverages supervision more explicit. This can help frame this method better and provide anchors to other colleagues in the community.

[Author Response · NeurIPS 2020]

Thank you for the detailed and constructive comments. Following the reviews, we conducted the following experiments:

1. We ran our method on RAVEN-FAIR [1], see the Fig. I. Note that some attributes are allowed to change when
no rules are applied on them. As noted by [1,B], the original RAVEN [A] is biased and CoPINet [13] exploits
this. Since CoPINet does not perform as well on unbiased data, we evaluated using MRNet (SOTA model with
80.6% accuracy) instead. The generation accuracy was 61.2%, this is to be compared to 66.8% on the target image
reconstructed by VAE, and only 9.0% on random generated image.
2. We conducted a user study, following the same scheme as the machine evaluation. We extensively trained three
participants on the task of PGM questions. After training, each got 30 random questions with the correct target
image (reconstructed) and 30 with the generated target instead. Human performance on the correct image was 72.2%,
and on the generated image was 63.3%. This small gap reassures that the generation is accurate.
3. A second user study was appropriate for untrained humans. This study is similar to the qualitative image analysis
shown in the paper (Fig. 4). In PGM, an image is correct if and only if it contains the right instance of the object
attribute which the rule is applied on (this information is in the metadata). By comparing the generated object
attribute to the correct image's object attribute – it can be easily determined if the generation is correct. The study
has 22 participants, 140 random image comparing instances for the generated answers, and 140 for a random choice
image (reconstructed by VAE) as a baseline. 70.1% of the generations were found to be correct and only 6.4% of the
random choice images (baseline). Considering that the SOTA model in the much easier task of recognition achieves
75.2% (MRNet), it seems that the generation accuracy does not fall much behind.
4. To demonstrate generalization, we trained on the "interpolation" regime of PGM in which the rules of train and test
differ. We evaluated using MRNet that was trained on the "neutral" regime (we measure the generalization of our
method, not of MRNet) and received 54.3% generation accuracy, this is to be compared to 70.8% accuracy on real
targets. This ability to generate abstract images based on patterns that are deliberately different than those of the
training set is quite remarkable.

**Reviewer 1:** We made the evaluation with multiple models simultaneously to not base the conclusion on a single model
that might be biased. Following the review, we added experiments with RAVEN-FAIR and two user studies (see points
1-3 above). As noted in point 1 above CoPINet's success is partly due to the ability to answer RAVEN questions without
looking at the context (the query).

**Reviewer 2: W1:** According to Sec.A.2 in the appendix of [10], human performance on PGM is very low. However,
very experienced participants scored well (80%). In the RAVEN dataset humans tend to score roughly 84% correctly on
average [A]. **W2:** We disagree that we just combine VAE and GAN in this work. VAE-GAN is used to an unconditional
generator, but this is only a starting point. In L147-151, we add a novel recognition path that selects a vector in the VAE
latent space. A novel relation-wise perceptual loss is defined (L173-196). These, in addition to the novel CEN model
in L100-103 that facilitates this, are the main contributions in this work. **W3:** As mentioned in L59-60, we use the
PGM [10] dataset. Following **R1**'s request we also use RAVEN-FAIR.

**Reviewer 3:** Untrained humans score low on PGM (see above); other PGM-based tasks also require supervised training.
This is similar to, e.g., computer chess. **Generation:** As we discuss, a multiple-choice protocol is easily exploitable. In
contrast, (1) generation is much more likely to require reasoning and (2) the important problem of abstract generation
was not studied in the past, as far as we can ascertain. **DS-KLD:** DS-KLD is designed to create variance in a subset of
the vector's indices. This subset is dynamic and depends on the nature of each vector. It could be beneficial for any
problem for which there are multiple modes to the latent representation and these modes are anisotropic. **User study:**
The 'blurry patterns' mentioned may not be a problem for a study if all choices are reconstructed by VAE. This way
users cannot pick on reconstruction artifacts, if these exist (this is why we apply such reconstruction in the user studies
in points 2+3 above). Note that humans without experience on PGM tests perform very poorly (see response to **R2**),
therefore it is hard to create a comprehensive study without investing considerably in training users. To circumvent this
limitation, we have also performed a user study that is suitable for untrained individuals. **Split:** We trained on the train
set and evaluated on the test set of PGM (and the same for RAVEN-FAIR). **out-of-distribution:** Following the remark,
we trained on the "interpolation" regime of PGM (see 4) and the results show very convincing generalization.

**Final note:** All reviewers have highlighted that the paper is well written, the idea is novel and interesting, agreed that
the results were good and the ablation study was extensive. We hope the reviewers would be able to read our rebuttal
with an open heart. Thank you.

Additional references:

[A] Zhang et al. RAVEN: A Dataset for Relational and Analogical Visual rEasoNing. CVPR 2019.

[B] Hu et al. Hierarchical Rule Induction Network for Abstract Visual Reasoning. arXiv 2002.06838, 2020.

Figure I: **Top:** A correct RAVEN-FAIR answer. **Bottom:** The generated answer (correct, even if differs, in green, incorrect in red).

[Meta-Review · NeurIPS 2020]

I have read the reviews and the author response and I have also asked an expert AC to also provide a comment in lieu of a 4th reviewer  (pasted below for reference). Taken all these together I will recommend acceptance, with a note.   NOTE TO AUTHORS: This work is going to be the reference paper for using generation as opposed to discrimination. As such, it is really crucial to set the right path for evaluating model in a fair and rigorous way, so that research that follows on builds on a solid base. The presented evaluation has some issues (see points bellow). Please, use the feedback provided and incorporate the human evaluation in the paper. ====== All reviewers agree that this is an intriguing way of viewing progressive matrices tests as a generation problem rather than a discrimination problem since. The authors themselves point as a motivation that "the ability to generate a correct answer is the ultimate test of understanding the question". I personally agree that this is an extremely interesting hypothesis, but the paper as it stands only goes half-way into question convincingly. Currently, the main evaluation of the paper seems to be focussing on the generation quality of a particular model itself, rather than whether or not the generation process provides intrinsically a better training signal than the discrimination one. At the same time, echoing R1, evaluating generation quality is currently problematic, since this is done using learned models, which to begin with are far from perfect, i.e., evaluating a perfect generation model would score anywhere from 75-85 depending on the underlying classification model. R2 pointed the need of human evaluation: The authors do provide some human results on the author response. My recommendation would be to try to clearly point that automatically evaluating generation is tough, present your results on that (I mean, you did the best you could) and also present the human evaluation. Perhaps the more interesting result is in Table 3. It is positive to see that the auxiliary classifier trained within this generation-pipeline improves upon other fully discriminative models. Shouldn't this also be somewhat central part of the evaluation for the models that will follow? On a final note, the authors claim in the discussion "Our work presents the first method to perform this task convincingly in the context of RPMs.", but this method doesn't really compare to baselines, so i don't think I even agree with this statement. Like, what would need to happen for the method to not perform convincingly? ======EXTRA REVIEW====== I think creating a neural network that can generate human-plausible answers to Raven's Progressive Matrices is a notable step forward in the list of things that neural networks (and ML in general) can do (given that the network is most definitely operating in the interface between symbolic / mathematical 'reasoning' and spatial/visual intelligence). I would have given the paper a score of 8 and after the author response, probably 9, and nominated for an honorable mention. The authors only consider one problem domain. A different domain would be the icing on the cake, but don't think it's critical because RPMs are very representative of visual-logical IQ tests in general. A main concern is that of the accuracy of the evaluation metric. The authors (during the rebuttal period) did a human evaluation which shows that humans generally choose the model's answer as the correct answer. This study should be included in the camera ready. Another solution would be to experiment with the 'interpolation' subset of the RPM dataset, where classifiers get 95% rather than 77% (because the problems involve less fine-grained distinctions in shades of grey). The paper could be written more clearly - it is very dense and I would have preferred a less symbolic/mathematical description of how the networks function and a more spatial/functional one. Overall, this is one of the most exciting papers I've seen in a long time. I strongly recommend acceptance.